# Reliability and Validity of a Novel Wearable Device for Measuring Elbow Strength

**DOI:** 10.3390/s20123412

**Published:** 2020-06-17

**Authors:** Marcus Brookshaw, Andrew Sexton, Chris A. McGibbon

**Affiliations:** 1Department of Mechanical Engineering, University of New Brunswick, Fredericton, NB E3B 5A3, Canada; drakemarcus@gmail.com; 2Institute of Biomedical Engineering, University of New Brunswick, Fredericton, NB E3B 5A3, Canada; asexton@unb.ca; 3Faculty of Kinesiology, University of New Brunswick, Fredericton, NB E3B 5A3, Canada

**Keywords:** isometric strength, portable strength measurement device, elbow flexors and extensors, repeatability and reproducibility, criterion standard validation

## Abstract

Muscle strength is an important clinical outcome in rehabilitation and sport medicine, but options are limited to expensive but accurate isokinetic dynamometry (IKD) or inexpensive but less accurate hand-held dynamometers (HHD). A wearable, self-stabilizing, limb strength measurement device (LSMD) was developed to fill the current gap in portable strength measurement devices. The purpose of this study was to evaluate the reliability and validity of the LSMD in healthy adults. Twenty healthy adults were recruited to attend two strength testing sessions where elbow flexor and extensor strength was measured with the LSMD, with HHD and with IKD in random order, by two raters. Outcomes were intra-rater repeatability, inter-rater reproducibility and inter-session reproducibility using intra-class correlation coefficients (ICC). Limits of agreement and weighted least products regression were used to test the validity of the LSMD relative to the criterion standard (IKD), and calibration formulas derived to improve measurement fidelity. ICC values for the LSMD were >0.90 for all measures of reliability and for both muscle groups, but over-predicted extensor strength and under-predicted flexor strength. Validity was established by transforming the data with the criterion standard-based calibration. These data indicate that the LSMD is reliable and conditionally valid for quantifying strength of elbow flexors and extensors in a healthy adult population.

## 1. Introduction

Strength measurements are key outcomes in surgical, clinical and academic research disciplines [1,2,3,4]. Valid, reliable strength measurements are essential not only to assessment outcomes [5,6], they often form the basis of clinical decision making in rehabilitation practice [7] and sport medicine [8]. However, options for reliable and accurate objective measurement of muscle strength in the clinic have not changed in three decade and remain limited to hand-held dynamometry (HHD) [9] and isokinetic dynamometry (IKD) [2], both of which have distinct advantages and disadvantages.

HHD is relatively inexpensive, highly portable, and provides an objective, scale measure of isometric strength. Although HHD is reported to be precise enough for clinical assessment [3,10], a detailed review of the literature shows the accuracy of the HHD is limited by the strength [11,12,13,14], stability [15,16], and expertise [17,18] of the rater. While IKD may overcome some of these limitations—indeed, it is considered the ‘gold standard’ for limb strength measurements [2,19,20,21,22,23,24] and has been shown to have high reliability [1,2,21,22,25,26,27,28]—it requires a specialized, stationary apparatus that is neither portable nor easily affordable, and requires technical expertise for operation and maintenance.

In this study we propose an alternative approach using a wearable limb strength measurement device (LSMD) [29] for objective strength assessment of elbow flexor and extensor muscle groups [30,31]. The device is described in Figure 1. The LSMD includes an aluminum frame with three members—a long bar with a spring-loaded selection knob at one end (inset Figure 1) connects to a wrist pad with an internal load cell (inset Figure 2) that aligns perpendicular to the wrist. The selection knob has spring-loaded pins that snap into a plate on a third bar with foam pads at either end. Using the knob, the user can select between the extension (Figure 1ii) or flexion (Figure 1iii) configurations for testing limb strength. Alternately, they can lock the device flat for storage/transport (Figure 1i). The portability and ease of use of the LSMD makes it appropriate for a broad spectrum of applications, including: athletics [17,32], physical therapy [33,34,35], planning and tracking medical interventions [18,36], and academic research [37]. However, the measurement performance of the LSMD has yet to be established.

The purpose of this study was: (1) to evaluate the reliability of the LSMD, in comparison to HHD and IKD performance, and (2) to evaluate the validity the LSMD using IKD as a ‘gold standard’, and develop correction coefficients for the LSMD to improve performance in both flexor and extensor muscle strength assessment.

## 2. Methods

Evaluation of the LSMD was accomplished using a two-stage experiment comparing isometric strength measurements of the human arm in flexion and extension using the LSMD, the HHD (MicroFET 2 hand-held dynamometer, Hoggan Scientific, LLC, Salt Lake City, UT, USA) and the IKD (Cybex Humac Norm, Computer Sports Medicine, Inc, Stoughton, MA, USA) configured for isometric strength measurement. Specifications for each device are shown below in Table 1.

The study was approved by the university research ethics board, and all participants provided informed signed consent prior to enrollment in the study.

### 2.1. Participants and Raters

Twenty healthy adults were recruited from the university student and staff population via poster advertisement and word of mouth. Included were adults between the age of 19 and 60. Exclusion criteria were upper limb fracture or elbow injury in the last 2 years, shoulder or elbow surgery in the last year, or stroke or other neurological disorder affecting the upper limbs.

In addition to the pool of healthy adult participants, a pair of raters was present for all tests, alternating between performing the tests and assisting the other rater, in accordance with the experimental design outlined below. Prior to the first round of testing, both raters received identical training in the use of the HHD, LSMD and IKD from qualified individuals. The two raters were denoted as rater A and rater B. Raters were blind to other raters’ assessments.

### 2.2. Experimental Design

A flow diagram of the experimental design is shown in Figure 3. Briefly, the design consisted of two sessions spaced 7–14 days apart. In session I, all 20 participants were assessed with all three devices by rater A. Order of device testing (HHD, LSMD, IKD) was randomized, as was the order of evaluating flexor and extensor strength. Three repetitions were conducted for each device and muscle group. This data was used to evaluate intra-rater reliability.

For session II, participants were randomized into two groups of 10 participants. Group 1 was assessed again by rater A, and group 2 was assessed by rater B. The former data was used to evaluate inter-session reliability, and the latter data used to evaluate inter-rater reliability, for each device and muscle group. Only the dominant arm was tested, determined by asking participants which hand they write with.

### 2.3. Protocol

All tests were conducted at the Andrew and Marjorie McCain Human Performance Lab at the University of New Brunswick. Standard protocols for HHD [40], IKD [41] and LSMD [30,31] were employed for isometric strength testing. Briefly, the elbow was palpated to locate the lateral epicondyle which was designated as the axis of rotation of the elbow joint. The location of the distal forearm load from the testing device was also measured to control for moment arm differences.

Figure 4 shows the positioning of the LSMD on a participant for flexor and extensor strength assessment. The LSMD fixes the joint at approximately 90 degrees flexion. Therefore, the isometric strength tests for HHD and IKD were also conducted at an elbow angle of 90 degrees. This is the typical testing angle for isometric strength measurement of elbow flexors and extensors [40].

Maximum voluntary isometric contractions (MVIC) were acquired for each device and muscle group three times each with a 30 s rest between repetitions. A minimum of 5 s rest was used between flexor and extensor tests, and a 20 min break was used between different device tests. The latter was required to ensure muscle recovery from the previous test, and to set-up testing with the other devices.

To minimize the impact of varied subject motivation and inter-subject variability in related psychological traits, a consistent, positive set of high demand instructions were used to motivate the users throughout their contractions [42].

### 2.4. Data Analysis

HHD and LSMD measure force, whereas the IKD measures torque. To facilitate comparison with the IKD, each MVIC recorded with the HHD and LSMD was converted to an equivalent torque at the elbow using the force measurements and the measured distances between the elbow joint center and line of action of the HHD and LSMD for each participant. As such, muscle strength was measured as a torque expressed in units of N*m.

#### 2.4.1. Objective 1: Reliability of LSMD for Strength Assessment

The torque data were used to determine two measures of reproducibility (inter-session and inter-rater reliability) and one measure of repeatability (intra-rater reliability), using the intraclass correlation coefficient ICC as described by Shrout and Fleiss [43].

For intra-rater reliability the three repeated measurements were treated as fixed effects in a two-way mixed effects ANOVA model (ICC model 3 for consistency, or ICC_con_(3,1)) to obtain the component of variance attributable to the targets (the cohort of 20 participants) and those associated with random error. The ICC_con_(3,1) was taken to be a measure of how well the same rater in identical conditions was able to consistently measure the strength of a participant.

For the two measures of reproducibility, each rater was assigned a different randomly selected subset of 10 participants from the full cohort—Group 1 was used to assess inter-session reliability and Group 2 was used to assess inter-rater reliability. Maximum strength from the three repetitions of each session was used in this analysis. ICC model 2 for absolute agreement, or ICC_aa_(2,1) using a two-way random effects ANOVA model was used to obtain the components of variance attributable to the targets (the cohort of 10 participants), judges (2 raters or 2 sessions) and those associated with random error. The ICC_aa_(2,1) was taken to be a measure of the agreement between two different raters and between two different sessions with the same rater.

In addition, a post-hoc analysis using simple main effects of the ANOVA model was used to follow-up on the significance of differences between raters and between sessions.

#### 2.4.2. Objective 2: Validity of the LSMD for Assessment

Using the torque data from Session I, the validity of the LSMD was evaluated by directly comparing it to the corresponding IKD data across all 20 participants. This comparison treated IKD as the ‘gold standard’ for isometric strength measurement, with the relative performance of the LSMD determined both qualitatively using limits of agreement (LoA) [44], and quantitatively, using weighted least products (WLP) regression [45]. The WLP regression was required rather than an ordinary least squares analysis due to the expected, and observed, heteroscedasticity of strength measures [46].

Although the LSMD was built with stiff steel members, the joint assembly and the foam contact rollers allow for small deflections of the device, especially with stronger users, potentially over- or under-estimating the elbow torque being applied. To correct for these effects, the WLP regression analysis between LSMD and the IKD from the above analysis was used to form the coefficients (slope *b*, and intercept *a*) of a linear regression equation
(1)$$\tau_{LSMD}=a+b*\tau_{IKD}$$
where $\tau_{LSMD}$ and $\tau_{IKD}$ are the measured torques from LSMD and IKD session I data, respectively. The coefficients can then be used to correct the measured LSMD data by calibrating the LSMD against the criterion standard IKD.
(2)$$\tau_{LSMD}^{Cal}=\frac{1}{b}\left(\tau_{LSMD}-a\right)$$
where $\tau_{LSMD}^{Cal}$ is the calibrated LSMD data.

Finally, to test the validity of the calibration, the LSMD data from Session II data were corrected with the derived equation and compared to Session II IKD data using WLP regression as described above.

All statistical analyses were conducted with SPSS (Statistical Package for the Social Sciences, v23, IBM-SPSS, Chicago, IL, USA).

## 3. Results

Twenty healthy adults (eight female) were recruited into the study. There were no drop-outs. Participant demographics and handedness, summarized by group and for the full cohort, are shown in Table 2. Mean peak elbow torques in extension and flexion for all three devices are summarized in Table 3 for each group and for the full cohort.

### 3.1. Reliability Analysis

#### 3.1.1. Intra-Rater Repeatability

This analysis used repeated measurement data (3 trials) from session I measured by rater A on the full cohort (*n* = 20). Reliability results using the ICC_con_(3,1) model are shown in Table 4. All three devices were found to be repeatable with all ICC values exceeding 0.96 except for flexion strength with IKD which was a little lower at 0.915 but still considered in the excellent range.

#### 3.1.2. Inter-Rater Reproducibility

This analysis used session I and session II data measured by rater A and rater B, respectively, on a sub-sample of participants (group 2, n_2_ = 10) measured by both raters. Reproducibility results using the ICC_aa_(2,1) model are shown in Table 4. ICC values for this analysis were lower than for intra-rater results. Although flexor assessment had good inter-rater reliability with all three devices with ICC values >0.85, for extensor strength measurement only the LSMD achieved good results (ICC = 0.897), whereas the HHD and IKD results were poor (ICC = 0.575 and 0.654 respectively).

#### 3.1.3. Inter-Session Reproducibility

This analysis used session I and session II data measured only by rater A, on a sub-sample of participants (group 1, *n*_1_ = 10) measured by this rater in both sessions. Reproducibility between sessions for a single rater using the ICC_aa_(2,1) model are shown in Table 4. Inter-session reproducibility results were excellent (ICC > 0.90) for all measures except flexion strength with the HHD, which had lower but good (ICC = 0.854) reproducibility.

#### 3.1.4. Post-Hoc Analysis of Reproducibility Results

Simple main effects were examined in the ICC_aa_(2,1) model for reproducibility across sessions (group 1) and raters (group 2). These data are shown in Table 5. In terms of between-session differences, only the HHD for flexor strength was statistically significant (*p* = 0.017). Between-rater differences were more severe with a significant difference for HHD in extension (*p* = 0.001), and IKD in both extension (*p* = 0.025) and flexion (*p* = 0.023).

### 3.2. Validity Analysis

#### 3.2.1. Criterion Validity

This analysis used session I data for the LSMD and the IKD devices. For the purpose of validation, the LSMD was considered the experimental measurement and IKD considered the criterion standard (i.e., ‘gold standard’) measurement. Model IIA regression results are shown in Figure 5.

The plots show that the LSMD over-predicts elbow extensor strength and under-predicts elbow flexor strength. Analysis of the regression coefficients for LSMD versus IKD showed a significant proportional bias effect for both flexion and extension, as shown in Table 6.

#### 3.2.2. Calibration Using the Criterion Standard

Regression coefficients from the above analysis of LSMD and IKD session I data were used to derive calibration formulas for LSMD elbow extensor and flexor measurement.
(3)$$\tau_{e}^{Cal}=\frac{1}{1.513}\left(\tau_{e}+9.022\right)$$
(4)$$\tau_{f}^{Cal}=\frac{1}{0.922}\left(\tau_{f}+1.459\right)$$
where *τ_e_* and *τ_f_* are the measured extensor and flexor torques, respectively, and *τ_e_^Cal^* and *τ_f_^Cal^* are the corrected torques.

Finally, Equations (3) and (4) were applied to LSMD data from session II and compared to IKD data for session II. Results are shown in terms of regression coefficients in Table 6 and measured torque values in Table 7. Before calibration, the LSMD overestimated extension strength by 21% and underestimated flexion strength by 14%. Calibration narrowed these effects, with the LSMD underestimating strength by 3% of the IKD measurement in both flexion and extension, falling within the random error measured during repeated measurements with the criterion device. LoA plots for extensor and flexor strength measurements are shown in Figure 6.

## 4. Discussion

Reliable and valid measurement of muscle strength is paramount for clinical decision making [5,6]. However, the literature shows few advances in strength assessment beyond the ubiquitous HHD and IKD devices introduced more than three decades ago. The LSMD was originally designed as a component of the BioTone™ system, intended for muscle tone assessment in the clinic of persons with upper motor neuron syndrome resulting from neurological injury or disease [47]. More broadly, the LSMD has potential for assessing elbow (and knee) strength in any population, for any number of compelling reasons. Whether quantifying strength of seniors for prescribing interventions, or evaluating contractile strength in athletes following sport-related injuries, the LSMD could provide a portable alternative to fixed-IKD. Furthermore, while the HHD is a clinically accepted approach to assessing strength, the superior reliability of the LSMD could improve the quality of strength assessments used in practice and in clinical trials of new treatments and drugs.

The purpose of this study was to evaluate the reliability and validity of the LSMD for these broader applications. This study focused on healthy adults as a first step to establish the quality, and limitations, of the measurement capability of the LSMD.

### 4.1. Reliability of the LSMD

To put the reliability results into context, recommended ICC cut-offs from the published literature were examined. Schrama et al. [3] provided a set of recommended cut-offs for ICC results in strength measurement studies that were specific to the type of reliability being measured. These cut-offs are shown below in Table 8. The guidelines of Lohr et al. [48] and Kottner et al. [49] were used to supplement the recommendations of Schrama et al. by providing a clinical viewpoint on minimal standards for ICCs. The analysis of van Trijffel et al. [50] provided a logical set of boundaries for judging the meta-reliability of a study that lists many ICC values associated with a single protocol.

While all three devices passed the van Trijffel et al. meta-reliability standards [50], the LSMD was the only device to exceed the ICC cut-offs in all testing conditions. In fact, the ICC values for the LSMD ranked highest for all but two of the six experimental conditions (see Table 4). It can be concluded that, despite any deficits in accuracy due to small deflections of the device, measurement of MVIC strength with the LSMD was repeatable and reproducible. This makes sense given that the LSMD structure (3-point contact) must equilibrate on the arm of the user arm during muscle contractions. Provided that the arm geometry of the user does not change between assessments, one would expect this equilibrium point to be consistent across assessments.

The IKD and the HHD devices also performed acceptably across each reliably metric. These data agree with the HHD reliability data presented in studies by Visser et al. [51], Bohannon [52] and Aufsesser et al. [53]. Similarly, the Cybex fixed-IKD reliability data agree with the intra- and rater reliabilities reported by Mathur et al. [54], Ekstrand et al. [55], and Stratford and Balsor [56].

The post-hoc analysis of inter-session and inter-rater reproducibility (Table 7) provided a better understand the reasons for observing lower reproducibility of the HHD and IKD for some measures. The between-raters data in Table 7 revealed a significant between-raters difference for the IKD in both extension and flexion. These data suggest that, for the Cybex IKD device, the source of the inter-rater variability is independent of the type of elbow motion. It can be concluded that alignment of the elbow joint in the Cybex machine was most likely responsible, given that it is the only source of variability directly involving the rater. While both raters were trained with the same instructional content and hands-on practice, clearly even small differences in alignment with the Cybex motor axis can result in less reproducible results, reported by others [21,57]. The ability of the LSMD to self-equilibrate likely explains why rater variability in aligning the user with the device was less influential.

While comparing reliability of the LSMD to the field ‘gold standard’ (IKD) is important, the reliability performance of the LSMD was also examined relative to the HDD. Although manual muscle tests are often the choice of test in the clinic [58], HHD is by far the most ubiquitous form of objective strength assessment in the clinic and research laboratory, and is known to be susceptible to variability across rater physical strength/size and skill level [10,34,59]. Both raters in this study were novice, though equally trained, assessors. As shown in Table 7, difference between-raters was observed for the HHD elbow extensor torque, and difference between sessions was noted for HHD elbow flexion torque.

The between-rater difference was likely due in part to a protocol deviation that occurred, reflecting the novice level of the two raters. One rater started with a break-test rather than a make test, which was then maintained for consistency. This would account for observed difference in elbow extension, since the ability to stabilize against strong is different between the make and break tests [56]. Finally it should be noted that a few participants in the study were, by all accounts, stronger than the raters, which no doubt contributed to the higher variability across raters and sessions [13,60].

### 4.2. Validity of the LSMD

Although excellent reliability was demonstrated for the LSMD, the validity of the device when tested against the criterion standard was less impressive before calibration. As shown in Table 3 and Figure 5, the LSMD over-predicted extensor muscle strength, and under-predicted flexor muscle strength. Fortunately, however, the disagreement between LSMD and criterion standard was a linear function of strength magnitude and was easily modelling using the WLP regression results. After applying these corrections to the measured LSMD data from a different session (data not used to create the calibration equations), the difference between LSMD and criterion standard disappeared, lending credence to this approach for improving the validity of both extensor and flexor muscle strength measurements.

There are some important implications of these findings. As noted above, the sample studied was from a population of university students and staff, representing a young and healthy population with higher than average strength compared to the general population (c.f. study by Hogrel et al. [61]) but demonstrating heteroscedastistic variability. Therefore, the effect of device deformation was driven by the strongest participants in the study. In clinical studies using the LSMD [30,31,62], participants were not likely strong enough to deform the LSMD. Nevertheless, the need for a correction at all remains a design concern.

To further explore this characteristic, Figure 7 shows a force diagram of the LSMD in its extensor and flexor configurations. Assuming the force through the wrist cuff remains perpendicular to the forearm, the force vectors through the proximal and distal upper arm cuffs provide insight into how the device self-stabilizes, and how it may in some circumstance negatively impact the measurement accuracy.

The right side of Figure 7 shows the device in extension strength configuration. Since the device represents a simple 3-force body, all three forces must pass through a common point in order for equilibrium to be achieved. For each configuration, vectors are shown at two extremes, where the green vectors show the desired force directions, and the orange vectors show the potentially problematic force directions.

For extension it can be seen that, depending on the direction of force through the posterior-proximal cuff, the force through the anterio-distal cuff located at the crux of the elbow could transmit loads to the forearm, thus inflating the measured elbow torque (shown by the orange vectors). For flexion it can also be seen that variability in anterio-proximal cuff force vector has a potential effect. If force directions are oriented as shown with the orange vectors, the net vertical force is directed upward, potentially causing the device to migrate upward on the arm and causing apparent under-prediction of flexor strength. It is likely that these biomechanical factors play a role in device measurement fidelity, and are currently the subject of further study and design by the authors.

### 4.3. Limitations

There are some important limitations of the study to point out. Although the full cohort was of sufficient sample size (*n* = 20), the sub-groups used to test reproducibility were smaller (*n* = 10), making them more susceptible to sampling bias. Nevertheless, ICC values agreed well with the literature and any aberrant findings were explainable.

The study only included healthy adult participants. Indeed, the intent of the device is for assessing strength across the population. This includes seniors and clinical populations that may have different underlying force generating characteristics, and thus reliability can not necessarily be extrapolated from the data in the present study. Future studies will be needed to evaluate the psychometrics of the LSMD in specific populations where applications exist. However, the data given in Figure 6 provides sufficient information for a clinician to infer the suitability of the LSMD for their individual needs.

This study only included the upper-extremity LSMD for the elbow joint. Although it is likely that the LSMD knee device (discussed in [30]) behaves similar to the elbow device, since it is essentially a scaled-up version of the arm model, future studies will need to apply the testing protocol to evaluate reliability and validity of the LSMD for lower-extremity assessment.

## 5. Conclusions

The reliability (repeatability and reproducibility) of the LSMD was found to be comparable to established commercial devices for measurement of elbow flexor and extensor strength in healthy adults. Validity of the LSMD was found to be acceptable when measurements were calibrated against the accepted gold standard IKD measurement. It is concluded that a calibrated wearable device such as the LSMD can be used to get reliable and valid measurements of elbow strength.

## Figures and Tables

**Figure 1 sensors-20-03412-f001:**
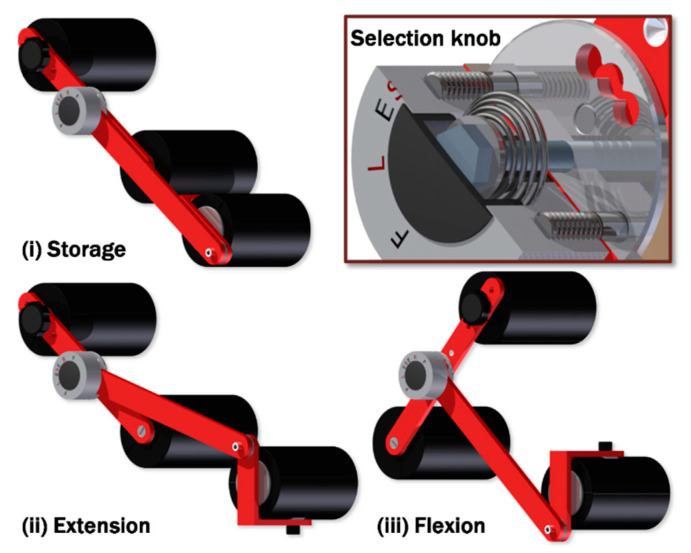
Limb strength measurement device (LSMD) in storage (**i**), extension (**ii**), and flexion (**iii**) configurations. *Inset:* Spring-loaded selection knob allows user to change between configurations.

**Figure 2 sensors-20-03412-f002:**
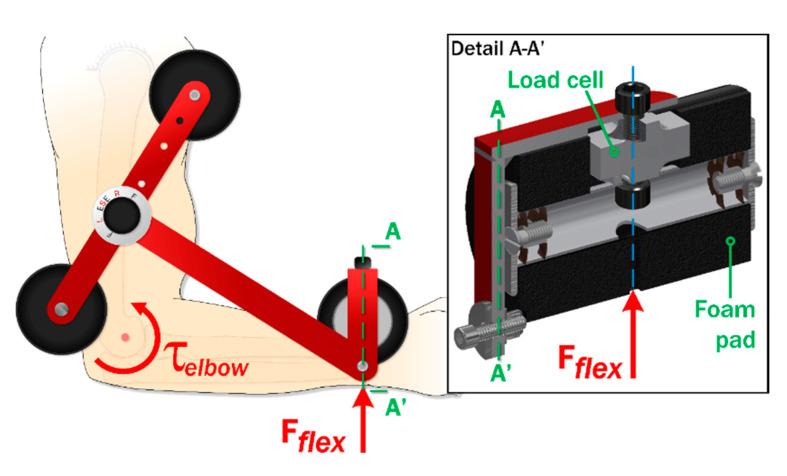
LSMD mode of operation—a wrist pad (Detail A-A’) with an integrated load cell is aligned with the wrist of the wearer, allowing the load cell to measure the flexion force (*F_flex_*) corresponding to the elbow torque (*τ_elbow_*) generated by the wearer. The identical principles apply in extension configuration.

**Figure 3 sensors-20-03412-f003:**
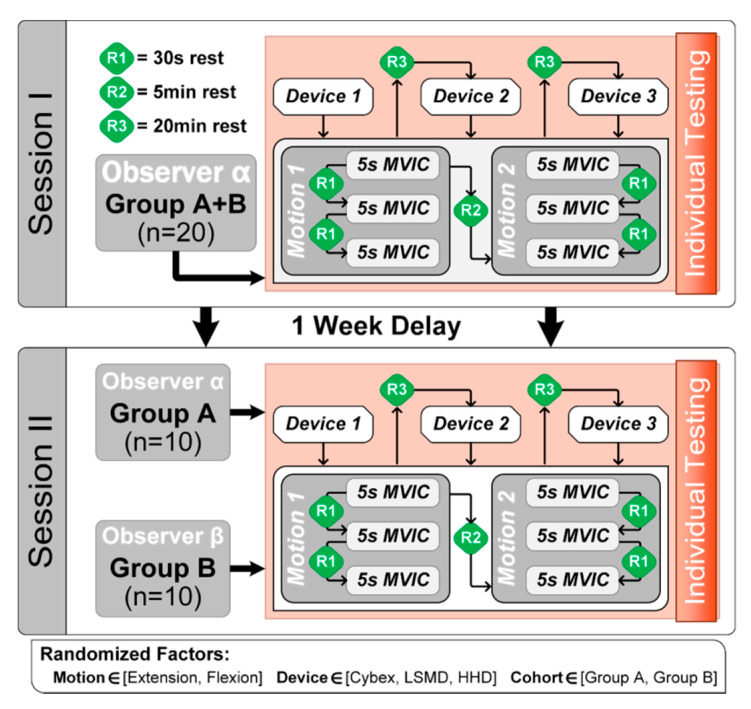
Flow chart of the experimental design.

**Figure 4 sensors-20-03412-f004:**
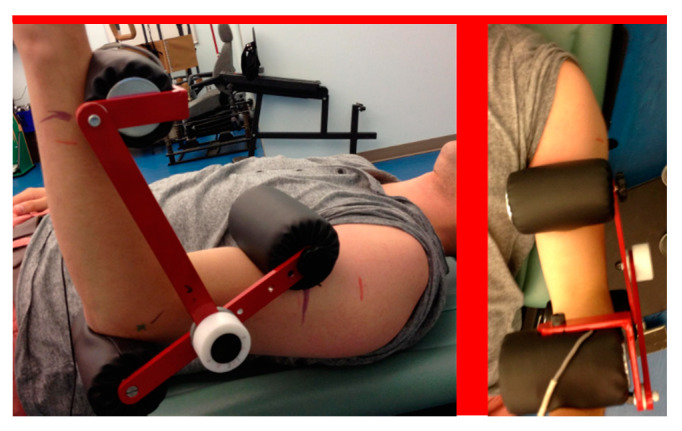
Testing set-up of LSMD in flexion (**left**) and extension (**right**).

**Figure 5 sensors-20-03412-f005:**
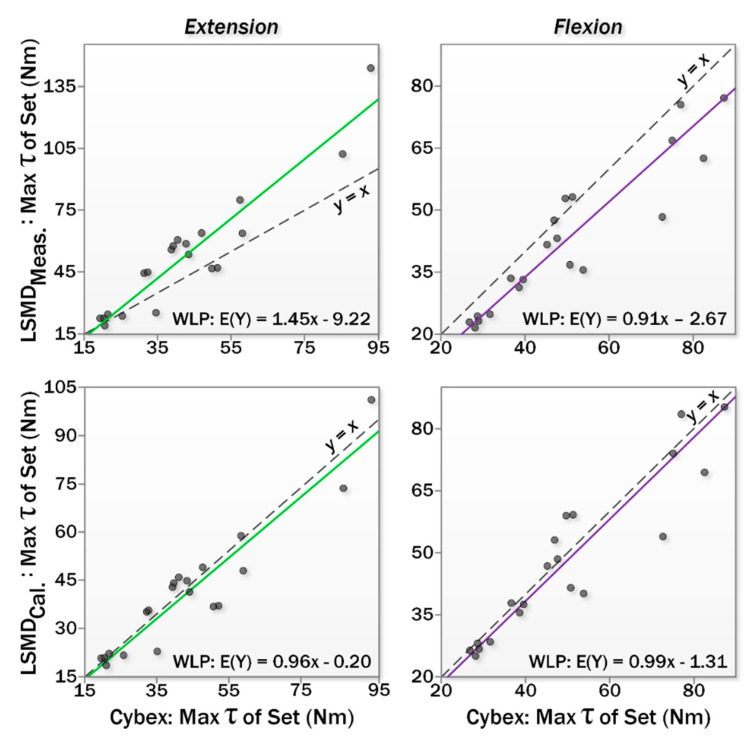
Weighted least squares regressions for LSMD versus IKD for extensors (**left**) and flexors (**right**), as measured (**top**) and after *post-hoc* calibration (**bottom**).

**Figure 6 sensors-20-03412-f006:**
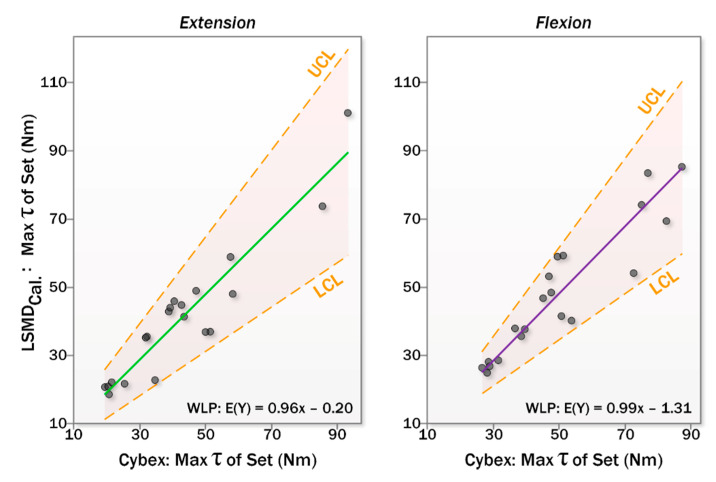
Limits of agreement plots for LSDM relative to IKD for extensors (**left**) and flexors (**right**).

**Figure 7 sensors-20-03412-f007:**
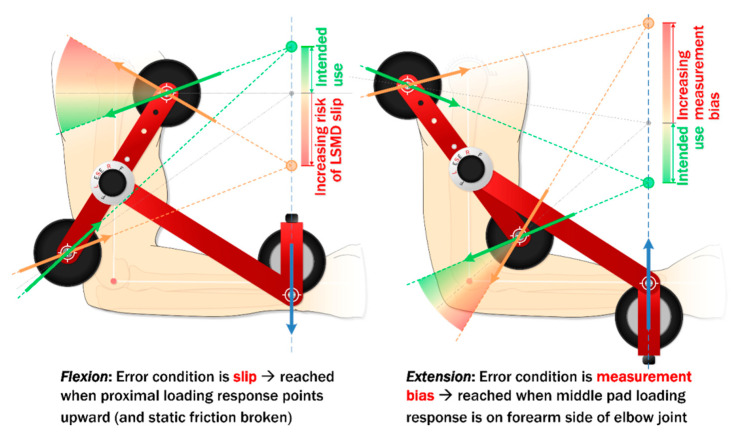
Biomechanical explanation of device measurement fidelity limitations.

**Table 1 sensors-20-03412-t001:** Comparison of specifications for LSMD, Cybex Isokinetic Dynamometer and MicroFET 2 Hand-held Dynamometer.

Device		Dimensions (cm)	Mass (kg)	Sensor	Power
**LSMD**	Arm	43 × 17 × 10 ^†^	1.45	Linear force: LC703–300 load cell (136 kg max capacity)	Internally powered: 9 V battery
	Leg	55 × 29 × 10 ^†^	2.64		
Cybex [38]		302 × 234 × 152	318	Axially aligned torque: Torque (678 Nm max) and angle (500°/s max)	Wall connection: Isolated, 20 A 220 VAC single-phase line
MicroFET 2 [39]		10 × 10 × 4	0.36	Linear force: Internal load cell (136 kg max capacity)	Internally powered: 3.7 V battery (½ AA cell)

^†^ In folded position.

**Table 2 sensors-20-03412-t002:** Participant demographics.

	Group 1	Group 2	Full Cohort
**Sex** n Female/Male	3 F/7 M	5 F/5 M	8 F/12 M
**Hand dominance** n Right/Left	8 R/2 L	8 R/2 L	16 R/4 L
**Age** mean years (s; min–max)	24 (3; 21–32)	28 (10; 21–53)	26 (8; 21–53)
**Height** mean cm (s; min–max)	171 (11; 152–188)	172 (11; 154–192)	172 (11; 152–192)
**Mass** mean kg (s; min–max)	73 (18; 48–101)	71 (16; 55–108)	72 (17; 48–108)
**BMI** mean BMI (s; min–max)	25 (4; 19–33)	24 (4; 19–33)	24 (4; 19–33)

*s* = Standard deviation; BMI = Body mass index; R = Right; L = Left; F = Female; M = Male.

**Table 3 sensors-20-03412-t003:** Peak torque characteristics of each participant group and full cohort.

		**Peak Extension *τ mean (s; max–min)***		
		**Group A**	**Group B**	**Full Cohort**
**HHD**	*S I*	47.1 (20.3; 20.6–78.7)	31.2 (7.9; 19.5–41.8)	39.2 (17.0; 19.5–78.7)
	*S II*	50.0 (25.5; 20.6–94.9)	40.0 (10.7; 25.0–56.0)	45.0 (19.7; 20.6–94.9)
**LSMD**	*S I*	59.0 (34.9; 19.7–120.2)	41.6 (17.7; 21.5–77.2)	50.3 (28.8; 19.7–120.2)
	*S II*	62.8 (39.0; 19.2–143.9)	43.4 (14.5; 23.7–65.1)	53.1 (30.3; 19.2–143.9)
**IKD**	*S I*	46.1 (22.6; 16.5–76.6)	31.8 (10.3; 18.3–54.5)	39.0 (18.6; 16.5–76.6)
	*S II*	47.3 (26.1; 19.5–93.0)	38.0 (10.4; 21.5–51.3)	42.6 (19.9; 19.5–93.0)
		**Peak Flexion *τ mean (s; max–min)***		
		**Group A**	**Group B**	**Full Cohort**
**HHD**	*S I*	44.4 (18.8; 16.6–75.5)	40.6 (10.5; 26.8–59.8)	42.5 (14.9; 16.6–75.5)
	*S II*	52.6 (23.1; 21.0–85.2)	38.8 (8.7; 27.2–54.3)	45.7 (18.4; 21.0–85.2)
**LSMD**	*S I*	45.0 (19.4; 20.8–68.8)	35.7 (10.3; 19.8–51.3)	40.3 (15.9; 19.8–68.8)
	*S II*	49.1 (21.1; 23.0–77.2)	36.6 (10.2; 21.7–52.9)	42.9 (17.4; 21.7–77.2)
**IKD**	*S I*	50.9 (23.2; 23.4–91.6)	40.3 (11.3; 24.0–60.4)	45.6 (18.6; 23.4–91.6)
	*S II*	55.3 (23.3; 26.7–87.2)	44.5 (13.3; 28.1–72.6)	49.9 (19.2; 26.7–87.2)

HHD = Hand-held dynamometry; LSMD = Limb strength measurement device; IKD = fixed-isokinetic (isometric) dynamometry; *s* = Standard deviation; *τ* = Torque in N*m; S I = Session one testing; S II = Session two testing.

**Table 4 sensors-20-03412-t004:** Reliability results for intra-rater repeatability, inter-rater reproducibility and inter-session reproducibility.

	Extension			Flexion		
		CI_95_			CI_95_	
	ICC	[LB, UB]	*d_Cohen_*	ICC	[LB, UB]	*d_Cohen_*
**Intra-rater Repeatability ^†^**						
**HHD**	0.975	[0.948, 0.989]	15.0	0.969	[0.935, 0.986]	12.9
**LSMD**	0.996	[0.991, 0.998]	34.6	0.965	[0.924, 0.985]	12.5
**IKD**	0.975	[0.947, 0.990]	15.0	0.915	[0.829, 0.964]	7.9
**Inter-rater Reproducibility ^‡^**						
**HHD**	0.575	[–0.107, 0.891]	2.7	0.875	[0.597, 0.967]	5.1
**LSMD**	0.897	[0.653, 0.973]	6.8	0.856	[0.526, 0.962]	5.6
**IKD**	0.654	[0.040, 0.904]	3.6	0.876	[0.397, 0.971]	6.9
**Inter-session Reproducibility ^‡^**						
**HHD**	0.909	[0.695, 0.976]	6.0	0.854	[0.297, 0.966]	5.2
**LSMD**	0.959	[0.852, 0.989]	11.3	0.938	[0.725, 0.985]	9.7
**IKD**	0.916	[0.700, 0.978]	7.5	0.933	[0.738, 0.983]	9.1

HHD = Hand-held dynamometry; LSMD = Limb strength measurement device; IKD = fixed-isokinetic (isometric) dynamometry; ICC = Intra-class correlation coefficient for consistency ^†^ (ICC_con_(3,1)) and absolute agreement ^‡^ (ICC_aa_(2,1)); CI_95_ = 95% confidence interval (lower bound, LB and upper bound, UB) on the ICC; *d_Cohen_* = Cohen’s d-index for effect size.

**Table 5 sensors-20-03412-t005:** Significance of the between-sessions and between-raters component of variance for ICC tests.

			Extension		Flexion	
n = 10 per Group			*F_j_* ^†^	*P_j_* ^†^	*F_j_* ^†^	*p_j_* ^†^
**Inter-session**	S I v. S II	HHD	0.9	0.372	8.4	0.017 *
**ICC_aa_(2, 1)**		LSMD	1.3	0.283	4.5	0.064
[Group 1]		IKD	0.1	0.721	3.3	0.104
**Inter-rater**	A v. B	HHD	24.5	0.001 *	1.4	0.260
**ICC_aa_(2, 1)**		LSMD	0.6	0.470	0.2	0.634
[Group 2]		IKD	7.3	0.025 *	7.4	0.023 *

HHD = Hand-held dynamometry; LSMD = Limb strength measurement device; IKD = fixed-isokinetic (isometric) dynamometry; ICC_aa_(2,1) = Intra-class correlation coefficient, model 2 for absolute agreement. S I = Session one testing; S II = Session two testing; A = Rater A; B = Rater B; ^†^
*F*-statistic and *p*-value for significance; ***** Significant effect with *α* = 0.05.

**Table 6 sensors-20-03412-t006:** Summary of WLP regression results for Session I and Session II.

Comparison			*r^2^*	*a*	CI_95_ for *a’*	*p_a_* ^†^	*b*	CI_95_ for *b*	*p_b_* ^†^
*Session I*									
***LSMD*** **vs.** ***IKD***		Ext.	0.810	−9.022	[−26.627, 8.583]	0.296	1.513	[0.870, 2.155]	<0.000 *
		Flex.	0.884	−1.459	[−8.005, 5.087]	0.645	0.922	[0.736, 1.108]	<0.000 *
*Session II*									
***LSMD*** **vs.** ***IKD***	***Measured***	Ext.	0.885	−9.220	[−21.973, 3.534]	0.146	1.451	[1.006, 1.895]	< 0.000 *
		Flex.	0.855	−2.666	[−10.856, 5.523]	0.503	0.912	[0.700, 1.124]	<0.000 *
	***Calibrated***	Ext.	0.885	−0.203	[−7.380, 6.974]	0.953	0.964	[0.732, 1.196]	<0.000 *
		Flex.	0.855	−1.315	[−8.841, 6.212]	0.718	0.990	[0.806, 1.174]	<0.000 *

LSMD = Limb strength measurement device; IKD = fixed-isokinetic (isometric) dynamometry; CI_95_ = 95% confidence interval on the indicated coefficient; ^†^
*p*-value for significance; ***** Significant effect with *α* = 0.05.

**Table 7 sensors-20-03412-t007:** Effect of calibration on session II LSMD torque data.

	Muscle	*τ* (Nm)		∆*τ* (IKD–LSMD) (Nm)			
	Group	IKD	LSMD	Mean	|min|	|max|	*s*
**As Measured**	Ext.	42.6	53.1	−10.4	1.5	50.9	13.4
	Flex.	49.9	42.9	7.0	0.7	24.2	7.2
**Calibrated**	Ext.	42.6	41.1	1.6	0.5	14.3	6.9
	Flex.	49.9	48.1	1.8	0.2	18.5	7.2

LSMD = Limb strength measurement device; IKD = fixed-isokinetic (isometric) dynamometry; *s* = Standard deviation; *τ* = Torque.

**Table 8 sensors-20-03412-t008:** Recommended cut-off values for ICC acceptability.

Reliability		Recommended Cut-Off	Source(s)
**Intra-rater**	ICC_con_(3, 1)	>0.90	[3,48,49]
**Inter-rater**	ICC_aa_(2, 1)	>0.75	[3,48]
**Inter-session**	ICC_aa_(2, 1)	>0.75	[3,48]
**Overall reliability of the study**		≥75% of listed ICC values are >0.75	[50]

Note: where more than one cut-off value was recommended in the literature, the most stringent value was chosen.

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
