# Peer review of "Reliability and Validity of a Novel Wearable Device for Measuring Elbow Strength"

_sensors, 2020, doi:10.3390/s20123412_

Round 1
Reviewer 1 Report
Overall, this manuscript is well-written and addresses a clear gap in assessing the limb strength measuring device that this group has developed. Please address the following comments prior to publication:
- Include a single, unambiguous reference in the Methods or Experimental Design section that readers can refer to for a detailed design/fabrication of the LSMD.
- Table 3: The labels for table 3a and 3b should be switched (i.e., Table 3a should be "Intra-rater reproducibility" and 3b should be "Inter-rater reproducibility."
- Discussion section, lines 274-279: The text in this section seems overly critical of the IKD/HHD inter-rater reproducibility given the documented deviation from the protocol. Given that only the extension measurements seem to be implicated (on line 302), it does not seem possible to provide an unbiased assessment of inter-rater reproducibility.
Reviewer 2 Report
The Authors report the assessment of a new wearable device for measuring elbow strength. The device has already been presented in previous works, here only a comparative study, with two other similar systems is presented. From this comparative study they claim that their device is reliable and conditionally valid for quantifying maximum voluntary isometric contractions in elbow flexors and extensors in a healthy adult population. In my opinion, several issues must be reviewed to make this work of interest to potential readers:
- Acronym section does not make sense when describing each of the acronyms in the following sections.
- A paragraph at the end of the Introduction section exposing the structure of the article is advisable.
- The paper flow is not smooth, technical writing is not up to the mark, and the article lacks structure. It is necessary to number the sections and subsections to better follow the manuscript. Some sections and subsections are too short, so they should be put together. The use of first persons (i.e., “we”, “their”, possessives, etc.) should be avoided, and can preferably be expressed by the passive voice or other ways. The combination of some images, diagrams and flowcharts in black and white and others in color produces little harmony. The axis titles and labels are oversize in Figure 5. Ideas are not well linked and overuse of parentheses is made. On the other hand, the equations do not follow the same format and in Equation 1 the multiplication sign should be used.
- Line 68: The device is described in Figure 1. Although the system has been presented in [30, 31], it would be suitable to provide more design details so that this article is self-contained. What is the selection knob in Figure 1? Another significant data would be the cost of this equipment compared to the IKD reference system, which is exposed as very expensive and currently inaccessible.
- It would be useful to provide images of the tests made with the other systems so that the reader can get an idea of the other two systems with which the comparison is made. A summary comparative table with the main characteristics of the three systems would be adequate to easily identify the performance of the systems under comparison.
- The conclusions are too short. This section should be completed and highlight the main findings of the work presented.

Round 2
Reviewer 2 Report
Although the presentation of the manuscript is better, I still think that the authors should improve it. Some examples are:
- The relationship of authors and their affiliations is much clearer if numbers are used instead of symbols.
- "Elbow flexors and extensors" and "Repeatability and reproducibility" are not keywords.
- The size of the tables must be reduced. Besides, the format and caption do not meet the criteria of the journal. The format of the references is also wrong.
- Unnecessary blanks on lines 220, 233, 354, 371, etc.
- The numbering of the equations should be on the same line as the equation.
- The font size and line spacing of lines 175-180 should be checked.
- The conclusions should go in a new section 5 and they should be objective, not subjective or vague as such sentences "While similar mechanics disadvantaged the LSMD in flexion, the effect was not substantial". As I mentioned in the previous review, the use of first persons (ie, "we", "their", possessives, etc.) should be avoided, and can preferably be expressed by the passive voice or other ways. “Our study shows that […]”.
What is most disturbing is the bibliography on which this work is based. Many references are 25 years old or more. I understand that some references with the foundations may be old and refer to these works in the first sections (Introduction or Methods, for instance) but not in the following. This is the case of references 41, 43, 44, 45, 47, 48, 49, 53, 54, 58, 60. To give an example, the manuscript makes claims of the following type "This is the typical testing angle for isometric strength measurement of elbow flexors and extensors [41]." 1986 reference. Does it make sense? I think more effort should be made to know the state of the art better and build on more recent works.
Finally, it is not apparent to me that the LSMD device is far less expensive than an IKD machine. The article should provide the information not expect the reader to assume.
